# Environmental and Biophysical Effects of the Bowen Ratio over Typical Farmland Ecosystems in the Loess Plateau

Xueyuan Ren [1], Qiang Zhang [1,2], Ping Yue [1,2,*], Jinhu Yang [3] and Sheng Wang [1,2]

1   Institute of Arid Meteorology, China Meteorological Administration, Lanzhou 730020, China;
    renxy@iamcma.cn (X.R.); zhangqiang@cma.gov.cn (Q.Z.); wangs@iamcma.cn (S.W.)
2   Key Laboratory of Arid Climatic Change and Reducing Deserter of Gansu Province,
    China Meteorological Administration, Lanzhou 730020, China
3   Lanzhou Regional Climate Center, Lanzhou 730020, China; yangjh@iamcma.cn
*   Correspondence: yuep@iamcma.cn

**Abstract:** The Bowen ratio ($\beta$) comprehensively reflects physical characteristics of the land-surface climate. In this study, eddy covariance systems installed at Dingxi and Qingyang were used to conduct energy distribution measurements and observations characteristic of semi-arid and semi-humid farmland ecosystems on the China Loess Plateau. We studied mechanisms by which eco-environmental factors influence $\beta$. Additionally, we investigated responses of physiological and ecological factors to water and heat exchange under seasonally dry and wet conditions within each farmland ecosystem. Our results showed that sensible heat flux in the semi-arid farmland was the main consumer of available energy. In the semi-humid area, latent heat flux in summer had the dominant role in energy distribution (mean $\beta$ 0.71). The $\beta$ in the semi-arid region was 1.5 times higher than that in the semi-humid region during the growing season. $\beta$ increased with an increase in the vapor pressure deficit (*VPD*) and ground–air temperature difference ($Ts - Ta$), and decreased significantly with an increase in effective precipitation and soil moisture. The change in $\beta$ with environmental factors was more clear-cut in semi-arid areas than in semi-humid areas. The Priestley–Taylor coefficient ($\alpha$) and $\beta$ satisfied a power function law in the growing season. There was a strong correlation between them, with the coefficients of determination for semi-humid and semi-arid areas being 0.62 and 0.72, respectively. $\beta$ decreased with an increase in the normalized difference vegetative index (*NDVI*), with this phenomenon being more obvious in the semi-humid zone ($R^2 = 0.40$). $\beta$ responded more rapidly to *NDVI* in the semi-arid area than in the semi-humid area. There was a negative exponential relationship between canopy stomatal conductance (*Gs*) and $\beta$, which displayed a stronger declining trend with the increase in *Gs* in the semi-arid area than in the semi-humid area. This study provides an important reference for the determination of land-surface characteristics of semi-arid and semi-humid farmland ecosystems on the Loess Plateau and for improving parameterization of land-surface processes.

**Keywords:** Loess Plateau; farmland ecosystem; physiological and ecological factors; Bowen ratio



## 1. Introduction

The farmland ecosystem is the foundation of the existence and development of human society, and represents an orderly structure composed of organisms in an environment that can realize the conversion of energy and matter [1]. In-depth research and scientific understanding of the influencing factors and regulatory mechanisms of farmland ecosystems can provide high-quality information that may potentially guarantee the sustainable development of society. Due to the interactions between biogeochemical cycles, climate, soil available water, and plant physiology, the distributions of sensible and latent heat fluxes in farmland ecosystems differ [2–4]. Studies have shown that climate change affects the energy change between the earth and the atmosphere through the water cycle [5–8]. In addition

to climate change, human activities (such as water conservancy projects and changes in land utilization) can also alter the water balance, thereby affecting the evapotranspiration process [5,9,10]. Meteorological and environmental factors, and the development of vegetation, can affect the distribution of surface energy during the growing season [11,12], in which the latent and sensible heat fluxes can change the environmental variables that affect matter and energy transfer between the atmosphere and the ecosystem. It has also been found that evapotranspiration in farmland ecosystems is mainly controlled by net radiation, but the regulation of latent heat transfers by vegetation indexes and canopy stomatal conductance ($Gs$) cannot be ignored [13–15]. In addition, vegetation phenology also affects the partitioning of net radiation to turbulent fluxes and soil heat flux [16–18].

The Bowen ratio ($\beta$) is a comprehensive physical index of the land surface climate, which comprehensively reflects the effects of microclimate and hydrological processes on ecosystem energy distribution and water use [19]. In previous studies of land–atmosphere interactions in ecosystems, $\beta$ has been found to be a very important factor [2,20]. However, due to changes in regional climate conditions (such as temperature, precipitation, and soil moisture) [12,21,22], and seasonal differences in the physiological characteristics of vegetation [13–15], there are often large differences in $\beta$ of ecosystems [23,24]. AmeriFlux observations have shown that the monthly average value of $\beta$ in farmland ecosystems is between 0.26 and 1.3 [25–27]. Even during a relatively stable growing season, there are still significant differences in $\beta$ among different farmland ecosystems [2,28]. For different ecosystems in the same climate region, there is an obvious contrast in their ability to regulate water and heat exchange, which is an internal factor leading to an apparent discrepancy in $\beta$. Precipitation is the most important driving factor in this process [2,24]. In the Loess Plateau, where precipitation fluctuates substantially, the ecological environment is fragile. The process of water and heat exchange in this region is also extremely sensitive to climate change [24], which makes $\beta$ more dependent on the driving effect of environmental factors. Therefore, studying the seasonal variation in $\beta$ of the typical farmland ecosystem of the Loess Plateau is of great significance for better understanding the land–atmosphere interaction mechanism in semi-arid regions.

The Loess Plateau in China is located within a typical semi-arid and semi-humid climate zone, which is not only a transitional zone for the East Asian summer monsoon, but is also positioned at the intersection of the water and heat gradient zones in China [25]. Therefore, the spatial distribution and temporal variation in land surface physical parameters in the Loess Plateau are very significant and highly sensitive to the advance and retreat of the monsoon and changes in its intensity. Due to the influence of the summer monsoon, the annual precipitation in this region is relatively concentrated, with about 65% of the total annual precipitation received from June to September. However, the interannual variability of precipitation is very large, which leads to visible spatial differences in the vegetation distribution [29,30]. The seasonal fluctuations of precipitation will undoubtedly lead to seasonal changes in $\beta$, which will cause the water and heat exchange of farmland ecosystems to display significant dry–wet conversion characteristics in turn [31,32]. The $\beta$ and its influence on grassland in the Loess Plateau have been studied in depth [24], and the evapotranspiration in farmland ecosystems and its environmental impact in this region are also well understood [33–35]. However, there have been few studies of water and heat exchange in farmland ecosystems on the Loess Plateau, especially in terms of $\beta$ and its influencing factors, despite it being a surface parameter that can comprehensively reflect the effects of water and heat. This has prevented the interactions between the land surface and the atmosphere in the farmland ecosystem of the Loess Plateau from being fully understood, and has prevented an in-depth understanding of water and heat exchange.

This study aimed to identify the effects of environmental factors on $\beta$ of farmland ecosystems in different climate regions of the Loess Plateau using experimental land–atmosphere interaction data for two typical farmland ecosystems in Dingxi and Qingyang, which are semi-arid and semi-humid regions, respectively. The remainder of this paper is organized as follows: The study area, data and method employed are described in Section 2.

The results of physiological and ecological factors relating to water and heat exchange are investigated briefly in Section 3. The discussion of the results is provided in Section 4. In Section 5, the conclusions of this paper are presented.

## 2. Materials and Methods

### 2.1. Site Description

Dingxi Station (35.58°N, 104.62°E) is located in the elevated extension area of the Loess Plateau, with an altitude of 1896.7 m. Precipitation from June to September accounts for 66% of total annual precipitation. The mean annual temperature and precipitation are 6.7 °C and 386 mm, respectively. The average annual pan evaporation is 1400 mm, and the annual mean sunshine duration of 2344 h is typical for a semi-arid climate. Qingyang Station (35.44°N, 107.38°E) is located in Dongzhiyuan on the Longdong Loess Plateau at an altitude of 1421 m and has an average annual temperature and precipitation of 8.8 °C and 562 mm, respectively. Precipitation from June to September accounts for 67% of the annual total. The average annual pan evaporation is 1470 mm, and the annual average sunshine duration of 2250 h is typical for a semi-humid climate. During the study, the principal crops in Dingxi were potatoes and spring wheat, whereas the principal crops in Qingyang were winter wheat and spring corn. The canopy height of the crops during the vigorous growth period was approximately 50 cm [34]. Both experimental sites are rain-fed farmland. Figure 1 shows the specific geographic locations.

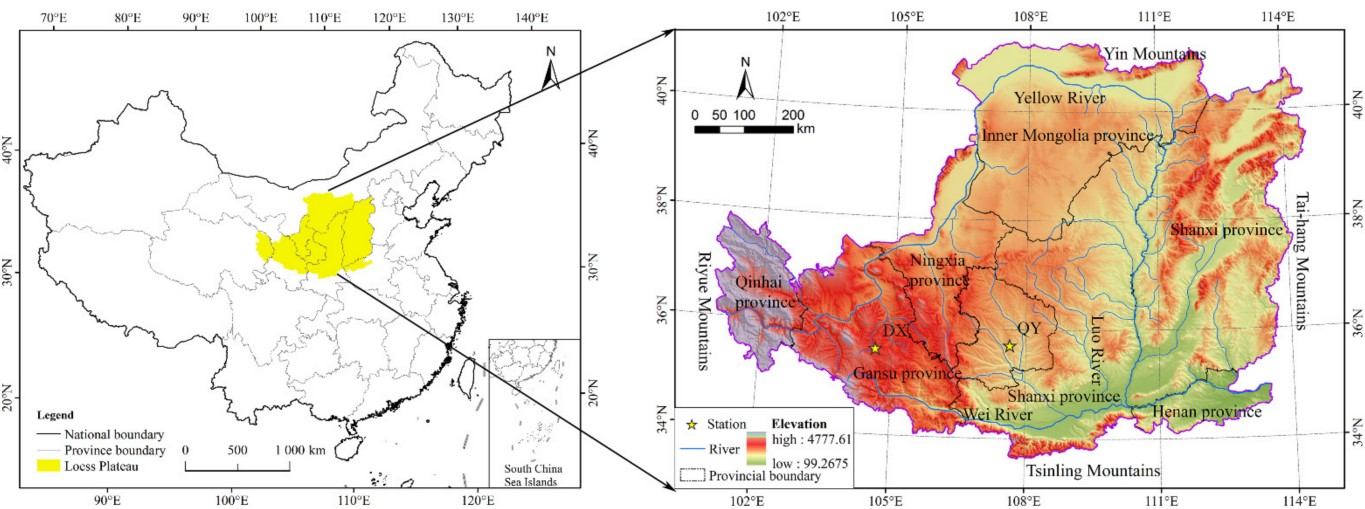

**Figure 1.** Geographical location of the study area. The stars show the Dingxi and Qingyang stations.

### 2.2. Observation Method and Data Processing

The data used in this paper include turbulence flux data observed by the eddy covariance system, temperature, humidity, and wind gradient data observed by a near-ground gradient tower, and radiation, soil temperature, and humidity gradient. Conventional observation data from meteorological stations were also used. The data period of turbulent flux in Dingxi is August 2016–May 2019; for Qingyang, multi-segment data were used, such as July 2011–July 2012, May 2013–October 2013, December 2015–May 2016, and May 2018–July 2019. In addition, the observation data of Dingxi and Xifeng meteorological stations from 1980 to 2010 were used. The installation height and details of the specific models of the measuring instruments are shown in Table 1.

**Table 1.** Measurement instruments and installation height.

| Instrument | Type | Installation Height | |
| --- | --- | --- | --- |
| | | **Dingxi** | **Qingyang** |
| Open Path $CO_2/H_2O$ Gas Analyzer | Li-7500, Li-Cor (Lincoln, NE, USA) | 2.5 m | 3 m |
| Three-dimensional (3D) sonic anemometer | CSAT-3, Campbell (Logan, UT, USA) | 2.5 m | 3 m |
| Temperature and relative humidity probe | HMP45C-L, Vaisala (Vantaa, Finland) | 1, 2, 4, 10, and 16 m | 2, 4, 8, and 18 m |
| Net radiometer | CNR4, Kipp and Zoned (Delft, The Netherlands) | 1.5 m | 1.5 m |
| Self-calibrating heat flux sensor | HFP01SC-L50, Hukseflux (Delft, The Netherlands) | 2, 5, and 10 cm | 1, 2.5 and 5 cm |
| Soil temperature profile sensor | STP01-L50, Hukseflux | 0, 5, 10, 20, 40, 50, and 80 cm | 0, 5, 10, 20, 40, 60, and 90 cm |
| Water content reflectometer | CS616-L, Campbell | 5, 10, 20, 40, 50, and 80 cm | 5, 10, 20, 40, 60, and 90 cm |

The turbulent flux data were processed by the EdiRe software (v1.5.0.32, Robert Clement, University of Edinburgh, Edinburgh, UK), which was developed by the University of Edinburgh for quality control and pre-processing. The operations included wild point removal, rotation coordinates, turbulence stationarity calculation, and water and $CO_2$ lag corrections. After quality control, the data were processed into 30 min average results. After excluding the outliers and precipitation period data, missing data for periods of less than 6 h were linearly interpolated, whereas missing data for periods of more than 6 h were interpolated by a look-up table method, which was based on the correlation between sensible heat, latent heat, net radiation, and water vapor pressure deficit (*VPD*) [36]. In addition, turbulent flux is greatly affected at night [37–40], and using midday data (09:00–15:00) can make the calculation results more reliable [2,41]. Beijing time was used in the study.

Due to the lack of station vegetation index observation data, the normalized difference vegetative index (*NDVI*) retrieved from the Aqua Moderate Resolution Imaging Spectroradiometer (MODIS, Phoenix, AZ, USA) data was used, with a temporal resolution of 16 days and a spatial resolution of 250 m (https://ladsweb.modaps.eosdis.nasa.gov/search/order/1/MOD13Q1--61). The *NDVI* of the experimental site was obtained from the average values of the four nearest grid points.

### 2.3. Energy Balance

The surface energy balance can be expressed as:

$$Rn = H + LE + G + S + Q \tag{1}$$

where *Rn* is the net radiation (W/m$^2$), *H* is the sensible heat flux (W/m$^2$), *LE* is the latent heat flux (W/m$^2$), *G* is the soil heat flux (W/m$^2$), *S* is canopy heat storage, and *Q* is the sum of all additional energy sources and sinks. Typically, *Q* is neglected as a small term. McCaughey [42] and Moore [43] suggested that canopy heat storage had a great effect on the degree of energy balance closure when the vegetation height was more than 8 m. Hence, the canopy heat storage term (*S*) was not taken into account in this study. The two principal methods for evaluating the degree of surface energy closure are the energy balance ratio (*EBR*) and the ordinary least squares (OLS) methods.

The EBR determines the degree of surface energy closure by calculating the ratio of turbulent flux to available energy during the study period:

$$EBR = \frac{\sum(H + LE)}{\sum(Rn - G)} \tag{2}$$

For *EBR* = 1, the surface energy is in an ideal equilibrium state. This method is ideal for evaluating the long-term energy closure state.

The *OLS* method is a simple regression model based on a hypothesis. It is based on the principle of the least squares method and is widely used in parameter estimation. When the sum of squares between the estimated value of the model and the experimental observation is at a minimum, the estimated value model is considered the optimal fitting model, and can describe the relationship between turbulent flux and available energy to the greatest extent. The slope of the regression model reflects the degree of surface energy closure. When the intercept of the regression curve is 0 and the slope is 1, the surface energy reaches the ideal closed state. Figure 2 shows the surface energy closure obtained by the *OLS* method. The black dotted line is the ideal state, and the grey shadow part (the slope ranges from 0.49 to 0.81) is the result reported by Li et al. [44] in evaluating the energy closure of flux observations of the terrestrial ecosystem in China. The slope calculated by the *OLS* method is closer to 1, indicating that the degree of surface energy closure is higher.

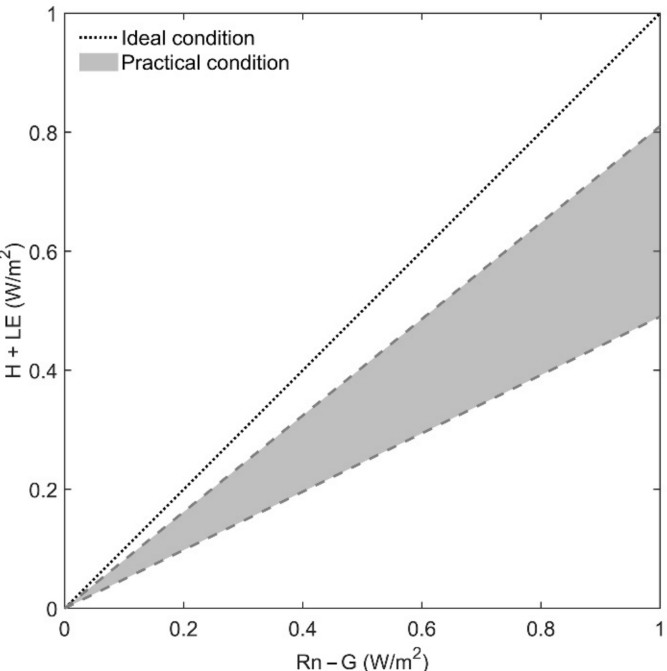

**Figure 2.** Surface energy closure obtained by the OLS method.

### 2.4. Soil Heat Flux Correction

The soil heat flux can be corrected to the surface value by the temperature integral method using the soil heat flux observed at 5 cm and soil temperatures at depths of 0 and 5 cm [45,46]:

$$G = G_5 + \frac{\rho_s c_s}{\Delta t} \sum_{z=5\text{cm}}^{z=0} [T((z_i, t) + \Delta t) - T(z_i, t)] \Delta z \tag{3}$$

where $G$ is the soil heat flux corrected to the surface (W/m$^2$); $G_5$ is the soil heat flux at 5 cm measured by the heat flux observation board (W/m$^2$); $T(z_i, t)$ is the soil temperature (°C) at depths of 0 and 5 cm; and $\rho_s c_s$ is the volumetric heat capacity of the soil, which was $1.24 \times 10^6$ J/(m$^3$·K) in the calculation. The soil temperature at 0 cm can be converted from surface long-wave radiation as follows:

$$T_0 = \left( \frac{R_L^\uparrow - (1 - \varepsilon_g) R_L^\downarrow}{\varepsilon_g \sigma} \right)^{1/4} \tag{4}$$

where $R_L^\uparrow$ and $R_L^\downarrow$ are the upward and downward long-wave radiation from the surface (W/m$^2$), respectively; $\varepsilon_g$ is the surface specific emissivity (0.96); and $\sigma$ is the Stefan–Boltzmann constant ($5.67 \times 10^{-8}$ W/$\left(\text{m}^2 \cdot \text{K}^2\right)$).

### 2.5. Bowen Ratio

The Bowen ratio ($\beta$) comprehensively reflects the impact of climate and hydrological processes on the energy distribution of land surface ecosystems and is expressed as the ratio of the sensible and latent heat fluxes:

$$\beta = \frac{H}{LE} \tag{5}$$

For $\beta > 1$, the sensible heat flux plays a dominant role in the energy distribution; for $\beta < 1$, the latent heat flux plays the leading role.

### 2.6. Overall Land Surface Parameters

Canopy resistance ($R_s$), dynamic resistance ($R_a$), and climate resistance ($R_i$) are important parameters that affect the study of land-atmosphere interaction [14]. $R_s$ is obtained by the Penman–Monteith equation [47]:

$$R_s = \frac{1}{Gs} = \frac{\rho c_p VPD + R_a LE(\Delta\beta - \gamma)}{\gamma LE} \tag{6}$$

where $Gs$ is stomatal conductance ($m/s$); $\rho$ is air density $\left(\text{kg/m}^3\right)$; $c_p$ is the specific heat of air $(1005$ J/$(\text{kg}\cdot\text{K}))$; $VPD$ is the saturated vapor pressure deficit (kPa); $LE$ is the latent heat flux (W/m$^2$); $\Delta$ is the slope of saturated vapor pressure curve (kPa/K); $\beta$ is the Bowen ratio; $\gamma$ is the dry and wet bulb constant (kPa/K); and $R_a$ is the aerodynamic impedance at the height of the canopy (s/m). $R_a$ can be calculated by the Monteith–Unsworth equation [48]:

$$R_a = \frac{u}{u_*^2} + 6.2u_*^{-0.67} \tag{7}$$

where $u$ is the wind speed at 2 m (m/s); and $u_*$ is the friction velocity (m/s). $\Delta$ can be calculated by the following formula:

$$\Delta = \frac{4098\left[0.6108exp\left(\frac{17.27T}{T+237.3}\right)\right]}{(T+237.3)^2} \tag{8}$$

where $T$ is the air temperature (K).

Climatic resistance $R_i$ reflects the degree of atmospheric demand for moisture under different surface available energy conditions [14]:

$$R_i = \frac{\rho c_p VPD}{\gamma(Rn - G)} \tag{9}$$

Using Equations (6), (7), and (9), it can be shown that $R_s$, $R_a$, and $R_i$ satisfy the following relationship:

$$\frac{R_s}{R_a} = k_0 + k\sqrt{\frac{R_i}{R_a}} \tag{10}$$

where $k_0$ and $k$ are empirical coefficients that depend on vegetation physiology and soil moisture status. For a clearer understanding of the impact of vegetation physiological processes on the water and heat exchange of the ecosystem, the normalized surface resistance

$Rs^*$ is defined to eliminate the difference in aerodynamic resistance and climate resistance caused by local changes in the underlying surface [23]:

$$R_s^* = \frac{R_s}{\sqrt{R_i R_a}} \tag{11}$$

In addition, the Priestley–Taylor coefficient ($\alpha$) can reflect the influence of environmental meteorological elements and vegetation physiological factors on ecosysterm evapotranspiration:

$$\alpha = \frac{LE}{LE_{eq}} \tag{12}$$

where $LE_{eq}$ (W/m$^2$) is the latent heat flux on a wide surface that is not restricted by moisture, defined as:

$$LE_{eq} = \frac{\Delta(Rn - G)}{\Delta + \gamma} \tag{13}$$

The value of $\alpha$ can be used to determine whether the evapotranspiration of the ecosystem is restricted by water conditions. When $\alpha < 1$, the evaporation of the ecosystem is limited by water. When $\alpha > 1.26$, there is no water stress in the ecosystem, and the factor affecting evaporation is only surface available energy ($Rn - G$) [49].

## 3. Results

### 3.1. Environmental Factor Variations

Figure 3 shows the seasonal variation in the characteristics of the environmental factors at Dingxi and Qingyang. The temperatures at the two stations had unimodal distributions (Figure 3(a1,b1)), reaching a maximum in midsummer and a minimum in January. The average temperature (*Ta*) at Dingxi was 0.2 °C higher than the 30 y (from 1988–2017) historical average (6.9 °C) during the experimental period. The monthly average minimum and maximum temperatures were −8.63 and 22.61 °C, respectively. *Ta* at Qingyang (10.2 °C) was 0.5 °C higher than the historical average; the average monthly minimum and maximum temperatures were −2.93 and 22.32 °C, respectively. During the experiment, maximum monthly precipitation in Dingxi and Qingyang was 148.1 (August 2017) and 246.2 mm (July 2018), respectively. Considering the monthly average precipitation over the past 30 years, the dry months at Dingxi station (April–October) accounted for 57.9% of the total; the dry months at Qingyang accounted for 35.0%. Due to the summer monsoon, more than 65% of the precipitation at the two stations was concentrated from July to September. Soil moisture greater than 40 cm in the tillage layer is very sensitive to precipitation. The monthly average *VPD* at Dingxi Station was 0.78 kPa and was largest in July 2017 (1.14 kPa) and smallest in January 2018 (0.21 kPa). The monthly average *VPD* at Qingyang Station was 0.69 kPa, with minimum and maximum values in November 2015 (0.13 kPa) and July 2019 (1.22 kPa), respectively. During the growing season, the average *VPD* values in Dingxi and Qingyang were 1.10 and 0.89 kPa, respectively. The seasonal variation patterns of the *NDVI* in Dingxi and Qingyang were basically the same. Vegetation growth and the *NDVI* increased in spring as precipitation and temperature gradually increased. The annual average values of the *NDVI* in Dingxi and Qingyang were 0.31 and 0.50, respectively; the respective growing season values were 0.40 and 0.56. The *Gs* trend was very similar to that of the *NDVI*, with an annual maximum from June to August. Precipitation was sufficient during the summer monsoon from June to August, and vegetation photosynthesis and transpiration were the strongest, which made the *Gs* reach the maximum. Vegetation physiological factors had the greatest impact on ecosystem evapotranspiration at this time, resulting in the peak value of $\alpha$ (Figure 3(a2,b2)).

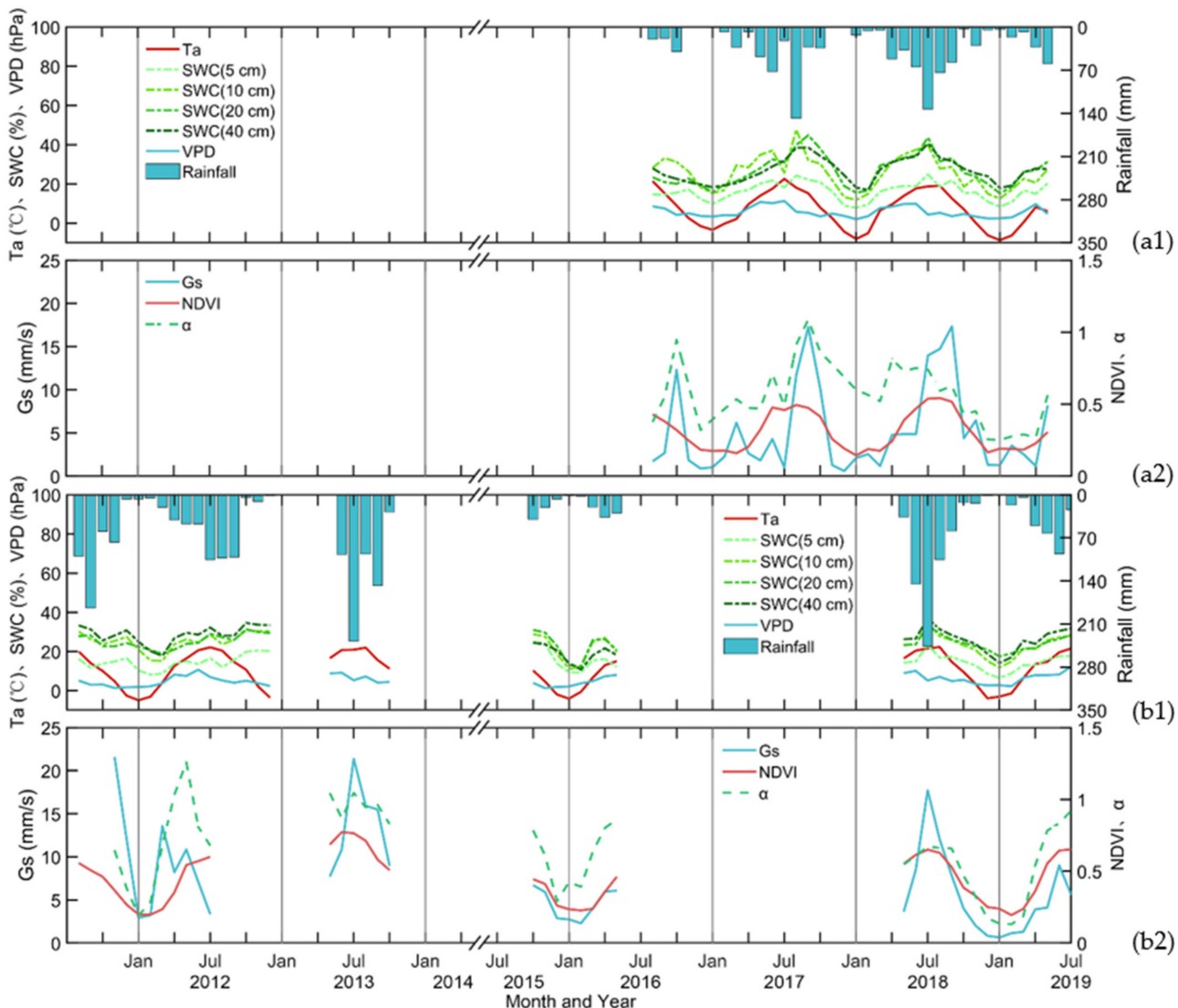

**Figure 3.** Seasonal and interannual variations of eco-environmental factors at Dingxi (**a1**,**a2**) and Qingyang (**b1**,**b2**) stations. *Ta* is the monthly mean temperature, *SWC* is the soil water content, *VPD* is the vapor pressure deficit, Rainfall represents monthly precipitation, *Gs* is the canopy stomatal conductance, α is the Priestley–Taylor coefficient, and *NDVI* is the normalized vegetation index.

### 3.2. Energy Balance Characteristics

The degree of energy closure is one of the most important criteria used to measure the quality and reliability of turbulent flux observation data [49]. The linear regression relationship between turbulent fluxes ($LE + H$) and available energy ($Rn − G$) is usually adopted to evaluate the reliability of eddy correlation system observations [50,51]. A large number of studies have shown that the energy imbalance observed by the eddy correlation method is between 10% and 30% [49,52]. Table 2 presents the energy closure of the Dingxi and Qingyang flux stations in the daytime, at night, and throughout the day. From the *OLS* results, the energy closure at both Dingxi and Qingyang was greater in daytime than that at night. This is due to the strong solar radiation during the day. The turbulent air is heated by the surface warming in the near-ground atmosphere, and the exchange is therefore greater [49,53], resulting in a high degree of energy closure. At night, due to the effect of stable atmosphere and low wind speed, the turbulent mixing is insufficient, resulting in low closing energy [12,54]. The increased uncertainty of net radiation measurement at night is also one of the reasons for the low energy closure [55]. When analyzing the degree of energy closure of flux observation systems, it has been reported that the energy balance

components of the land surface are not in the same physical measurement plane, which is also the objective reason for the phenomenon of energy non-closure [56].

**Table 2.** Characteristics of the energy balance in different regions.

| Site | Midday | | | | Night | | | | All Day | | | |
|---|---|---|---|---|---|---|---|---|---|---|---|---|
| | Sample Numbers | OLS | | EBR | Sample Numbers | OLS | | EBR | Sample Numbers | OLS | | EBR |
| | | Slope | $R^2$ | | | Slope | $R^2$ | | | Slope | $R^2$ | |
| Dingxi | 10,516 | 0.65 | 0.68 | 0.89 | 11,190 | 0.18 | 0.05 | 0.03 | 40,350 | 0.76 | 0.81 | 0.68 |
| Qingyang | 5149 | 0.71 | 0.71 | 0.81 | 7773 | 0.11 | 0.05 | 0.49 | 24,064 | 0.73 | 0.85 | 0.60 |

The *OLS* method commonly used in energy balance analyses is a simplified processing method based on the assumption of "no random error". The residual frequency distribution can be used to determine whether the model satisfies the hypothesis. Figure 4 shows the residual frequency distribution of the research site during the daytime. It can be seen that both residual density curves followed a normal distribution, indicating that the linear model obtained by the *OLS* method satisfies the assumption of "no random error". For observation data with a longer time scale, the *EBR* can balance the influence of error on energy closure. A large number of studies have shown that the energy closure calculated by this method is generally higher than that calculated by the *OLS* method [12]. It can be seen from Table 2 that the magnitude of the whole-day energy closure of Dingxi and Qingyang stations was between that of day and night, i.e., the closure of surface energy in daytime is greater than that during the whole day and at night. Compared with Li et al. [44], who used the *OLS* (0.49–0.81) and *EBR* (0.58–1.00) methods to evaluate the energy closure of ChinaFlux sites, the energy closure of the research site used in this study was slightly higher. This indicates that the accuracy of the observation data was high, and was suitable for a study of water and heat exchange in farmland ecosystems.

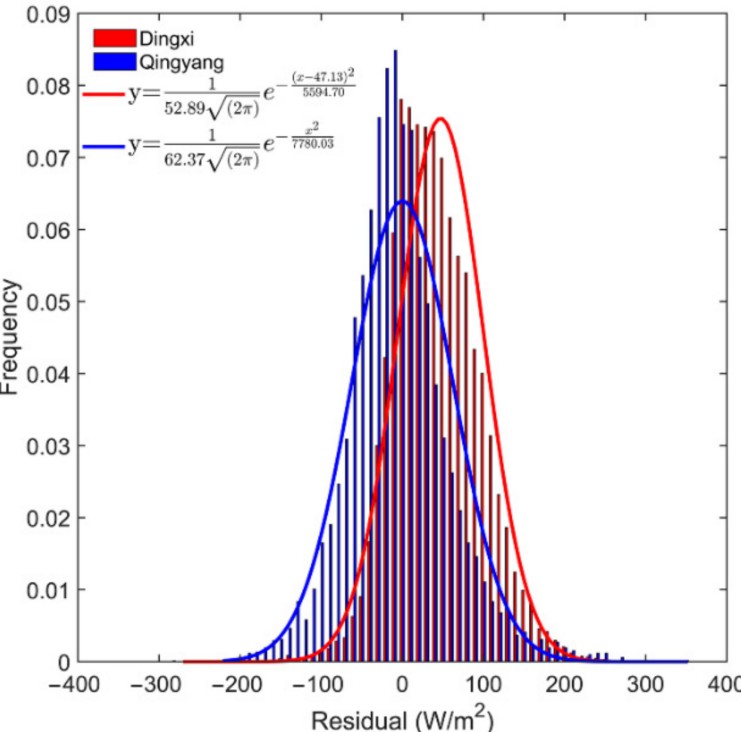

**Figure 4.** Frequency distribution and probability density curve of the energy balance residual.

### 3.3. Diurnal Cycle and Seasonal Variation in Energy Flux

When the annual average daily change in energy flux (Figure 5) was considered, the difference in the daily peak of net radiation between Dingxi and Qingyang was only 14.3 W/m² (Table 3). The daily peak values of the sensible heat flux were almost the same (120.3 and 122.3 W/m² in Dingxi and Qingyang, respectively). However, the difference in the latent heat flux between the two locations was large. The daily peak value of the latent heat flux in Qingyang in the semi-humid climate region was almost twice than that in Dingxi in the semi-arid climate region. The daily peak soil heat flux in Qingyang was approximately 2/3 of the Dingxi value. In addition, the latent heat flux (20.95 W/m²) in the arid region of the Loess Plateau was less than the sensible heat flux (28.98 W/m²), whereas the latent heat flux in the semi-humid region was larger than the sensible heat flux, with values of 41.41 and 28.50 W/m², respectively. According to the ratio of turbulent flux to net radiation (Figure 5e,f), the sensible heat flux in Dingxi was higher than the latent heat flux, with an average difference of 10.3%. In Qingyang, on the contrary, the average difference between the proportion of latent heat flux and the proportion of sensible heat flux was 7.6%. As a consequence, the sensible heat flux played a dominant role in the energy distribution in the semi-arid region of the Loess Plateau, whereas the latent heat flux was dominant in the semi-humid region.

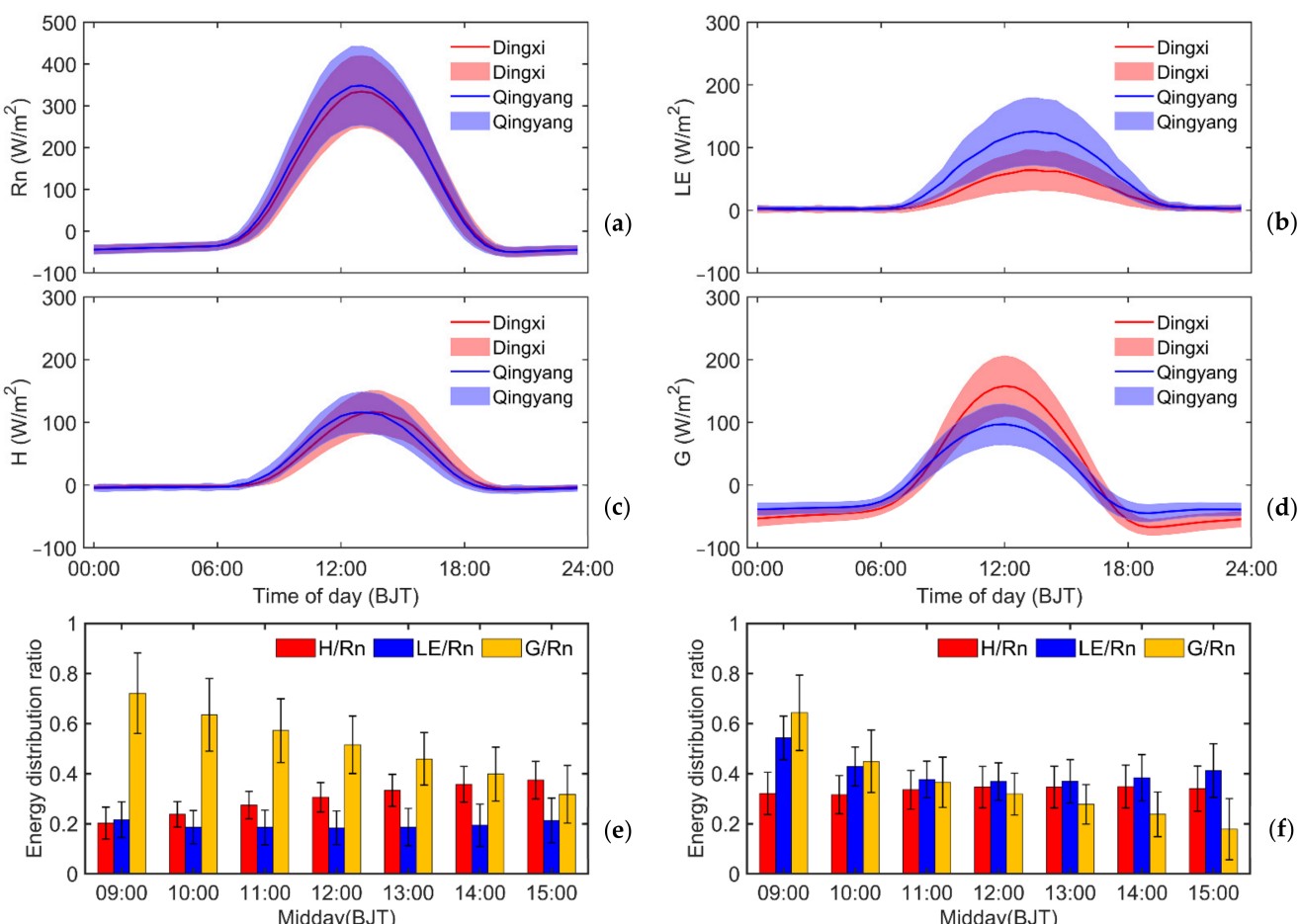

**Figure 5.** Annual average diurnal variation in energy flux (**a**–**d**), and the ratio of *H*, *LE*, and *G* to *Rn* at midday at Dingxi (**e**) and Qingyang (**f**). *Rn* is net radiation, *H* is the sensible heat flux, *LE* is the latent heat flux, and *G* is the soil heat flux.

**Table 3.** Annual average daily peak values and average daily values of the energy components.

| Site | $Rn$ (W/m$^2$) | | $LE$ (W/m$^2$) | | $H$ (W/m$^2$) | | $G$ (W/m$^2$) | |
|---|---|---|---|---|---|---|---|---|
| | Peak | Mean | Peak | Mean | Peak | Mean | Peak | Mean |
| Dingxi | 334.10 | 67.29 | 64.14 | 20.95 | 120.29 | 28.98 | 157.88 | 6.08 |
| Qingyang | 348.37 | 71.74 | 125.83 | 41.41 | 122.32 | 28.50 | 96.93 | 1.62 |

Figure 6 shows the daily distribution characteristics of the energy flux on a seasonal scale. The net radiation in the semi-arid and semi-humid regions of the Loess Plateau (represented by Dingxi and Qingyang) reached a maximum in summer, with daily average radiation intensities of 109.9 and 119.8 W/m$^2$, respectively, and a minimum in winter, with average values of 25.3 and 26.2 W/m$^2$, respectively (Table 4). The daily peak values of net radiation in Dingxi and Qingyang occurred at 13:00, with summer values of 440.2 and 460.22 W/m$^2$ and winter values of 226.52 and 240.9 W/m$^2$, respectively. In summer and autumn, the latent heat flux at Qingyang in the semi-humid area accounted for 61.8% and 77.7% of the net radiation, respectively; whereas the sensible heat flux accounted for 25.5% and 33.8% of the net radiation, respectively. The latent heat flux at Dingxi in the semi-arid area in summer and autumn accounted for 32.9% and 41.3% of the net radiation, respectively; whereas the sensible heat flux accounted for 37.8% and 36.3%, respectively. The maximum value of soil heat flux at Dingxi and Qingyang appeared in summer, 125.82 W/m$^2$ and 214.49 W/m$^2$, respectively. Compared with other energy components, soil heat flux has a larger nighttime variability. This is related to two factors: the change in the direction of soil heat transfer caused by the process of soil freezing and thawing [57]; and the different energy intensity of soil radiation to the atmosphere caused by the diurnal variation in ground temperature difference in different seasons. Zhang et al. [58] reported a similar conclusion in the Loess Plateau.

Figure 7 shows the seasonal variation in energy flux in Dingxi and Qingyang. The net radiation in the semi-arid and semi-humid areas of the Loess Plateau had single-peak distributions, with maximum values in July (111.2 W/m$^2$) and June (125.0 W/m$^2$), respectively, and minimum values in December (21.6 and 19.5 W/m$^2$, respectively). Sensible heat flux and latent heat flux are not only restricted by net radiation, but are also affected by surface vegetation and soil moisture. Dingxi and Qingyang are bare land in the non-growing season, where precipitation is less than 20% of the annual total. Therefore, whether it is a semi-arid or semi-humid area of the Loess Plateau, the ratio of sensible heat flux to net radiation is relatively large (Figure 7(a1,a2)). For the semi-arid area of the Loess Plateau, with the increase in net radiation from March to May, and under the constraint of water conditions, net radiation is mainly transformed into sensible heat flux. Nevertheless, as the summer monsoon advances, the region that is located at the northern edge of the typical summer monsoon transition zone is affected by monsoon precipitation; as a result, the latent heat flux from June to September is generally equivalent to the sensible heat flux (Figure 7(a2)). However, due to the large fluctuation in monsoon precipitation, this area often experiences the phenomenon in which sensible heat and latent heat flux alternately dominate the energy distribution. In contrast, the unimodal distribution of the latent heat flux in Qingyang in the semi-humid region of the Loess Plateau was more prominent than that in Dingxi in the semi-arid region, with a peak value of 77.9 W/m$^2$ in July. The experimental results showed that the latent heat flux in the growing season in the semi-humid region of the Loess Plateau was 2.4 times than that of the sensible heat flux. The average latent and sensible heat fluxes were 69.4 and 29.1 W/m$^2$, respectively. In the same period, the latent and sensible heat fluxes in Dingxi were similar (33.1 and 39.4 W/m$^2$, respectively).

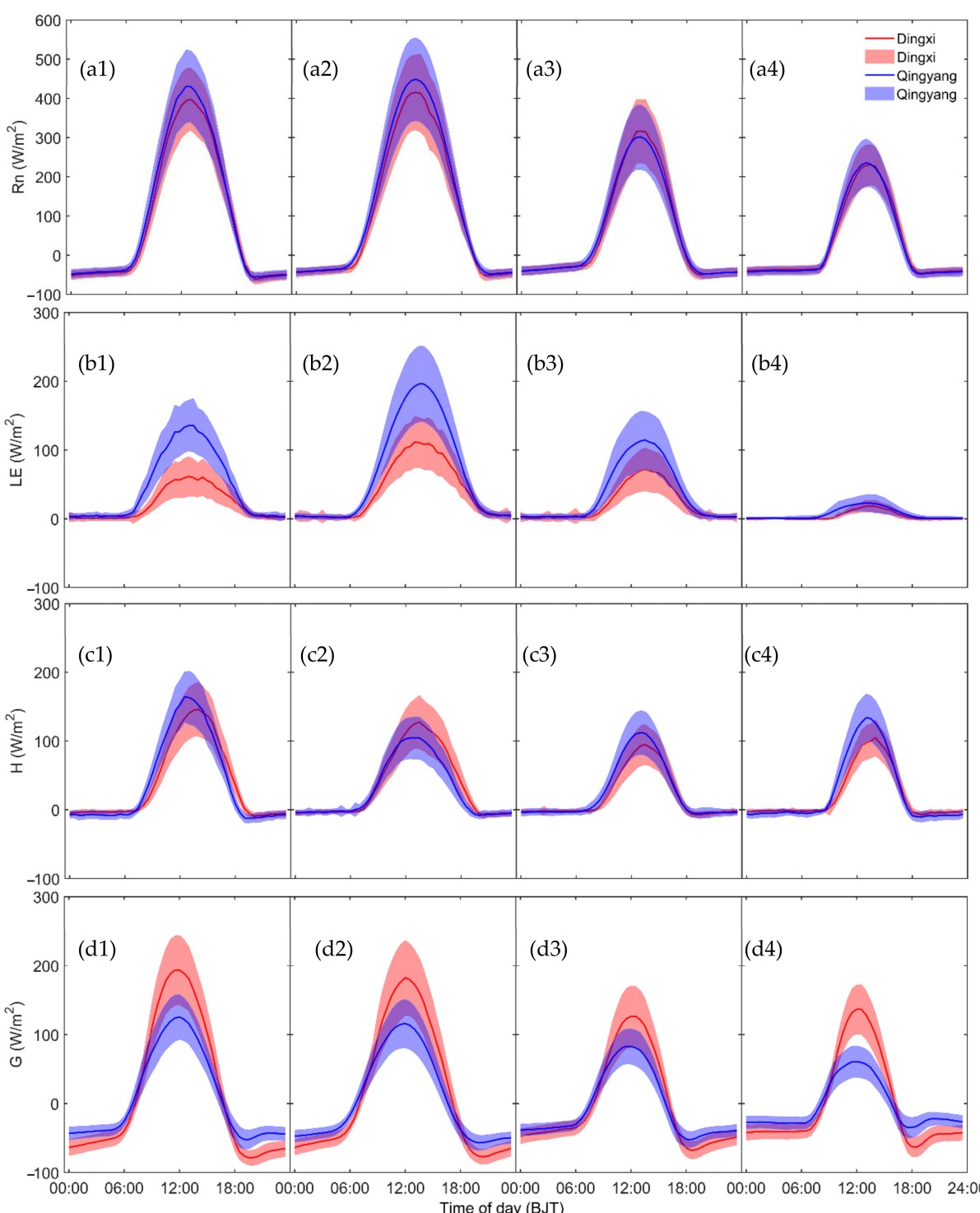

**Figure 6.** Seasonal average diurnal variation in the energy flux (**a–d**). Columns 1, 2, 3, and 4 represent spring, summer, autumn, and winter, respectively.

**Table 4.** Peak and daily average values of the seasonal average daily energy fluxes.

| Energy Component (W/m²) | | Spring | | Summer | | Autumn | | Winter | |
|---|---|---|---|---|---|---|---|---|---|
| | | Dingxi | Qingyang | Dingxi | Qingyang | Dingxi | Qingyang | Dingxi | Qingyang |
| $Rn$ | Peak | 345.02 | 376.71 | 440.16 | 460.16 | 362.38 | 348.03 | 226.45 | 240.88 |
| | Mean | 67.73 | 77.14 | 109.87 | 119.84 | 78.76 | 75.02 | 25.34 | 26.20 |
| $LE$ | Peak | 39.31 | 82.61 | 103.10 | 169.41 | 101.19 | 168.67 | 22.55 | 21.63 |
| | Mean | 11.77 | 25.70 | 36.00 | 74.11 | 32.54 | 58.31 | 5.23 | 6.64 |
| $H$ | Peak | 148.75 | 162.54 | 143.10 | 112.81 | 107.21 | 110.68 | 93.27 | 118.66 |
| | Mean | 34.85 | 34.02 | 41.31 | 30.49 | 28.59 | 25.33 | 17.93 | 21.93 |
| $G$ | Peak | 173.24 | 109.18 | 214.49 | 125.82 | 124.15 | 97.69 | 136.16 | 56.19 |
| | Mean | 10.34 | 6.68 | 20.07 | 8.67 | 1.24 | −2.29 | −3.29 | −6.94 |

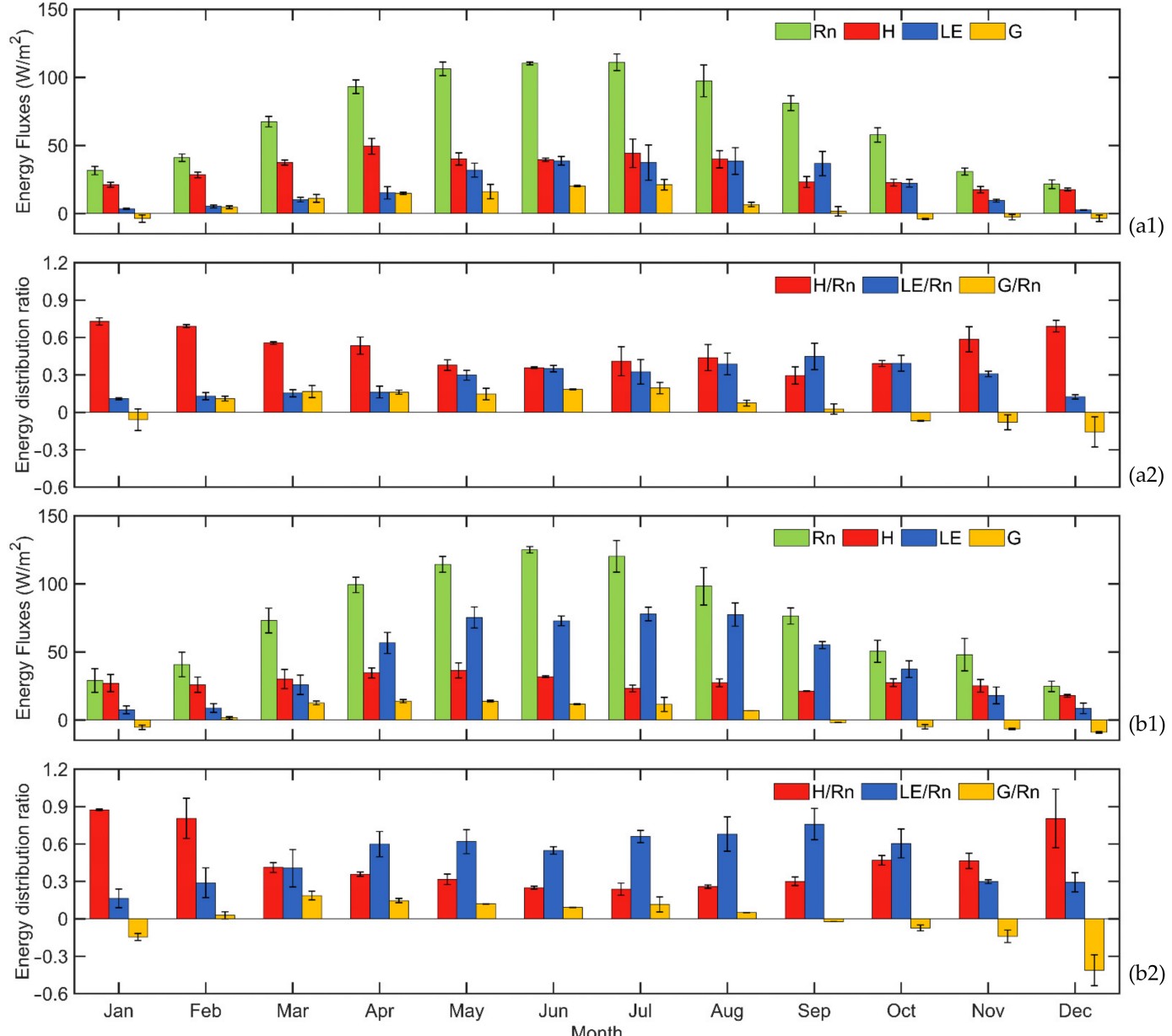

**Figure 7.** Seasonal variation in the energy components (**a1**,**a2**) and the ratio of *H*, *LE*, and *G* to *Rn* in Dingxi (**b1**) and Qingyang (**b2**).

In addition, the cumulative fraction curves of the sensible heat flux in Qingyang during the whole year and the growing season basically overlapped, when the cumulative fraction reached 0.6 (Figure 8a). When the cumulative fraction was about 0.4, the annual sensible heat flux was basically consistent with the latent heat flux in Dingxi. The cumulative fraction curves of the sensible and latent heat fluxes in the growing season in the two regions show that the sensible and latent heat fluxes in Dingxi in the semi-arid region were basically the same for cumulative fractions > 0.6, whereas the sensible heat flux in Qingyang in the semi-humid region was almost half of the latent heat flux. These results were consistent with the fluctuation in the sensible and latent heat fluxes in Figure 8b. The average latent heat flux (31.2 W/m$^2$) was less than the average sensible heat flux (34.1 W/m$^2$) in the semi-arid area during the growing season; contrary results were found for the semi-humid area. The interquartile ranges of sensible and latent heat fluxes in semi-arid regions are 23.27–43.42 W/m$^2$ and 21.00–42.74 W/m$^2$, respectively, and in sub-humid areas the ranges are 22.46–36.72 W/m$^2$ and 42.45–70.07 W/m$^2$, respectively. Zhang et al. [58] also showed that the summer latent heat and sensible heat flux in the semi-arid area of Northwest China are equivalent, and the summer latent heat flux in the semi-humid area is about twice the sensible heat flux.

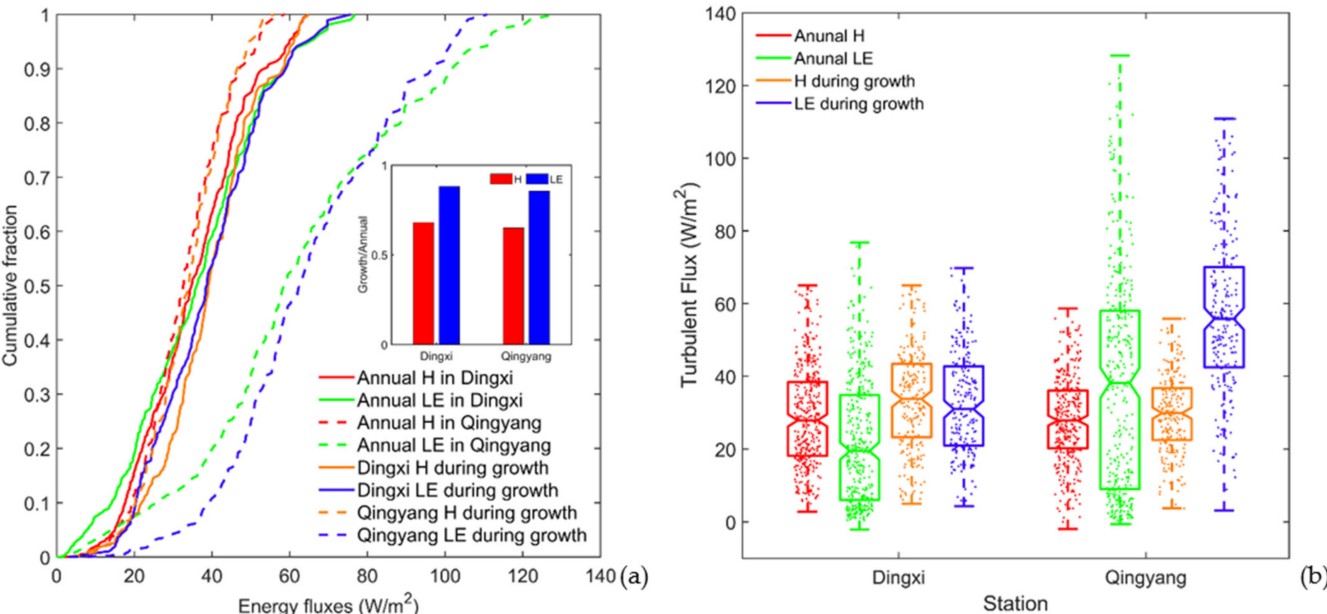

**Figure 8.** Cumulative fraction curve (**a**) and box plot (**b**) of the sensible and latent heat fluxes in Dingxi and Qingyang.

The soil heat flux changed from negative to positive from January to February, and from positive to negative from August to September (Figure 7(a2,b2)). This shows that the conversion of the heat source and heat sink occurred in the soil during these two periods. The soil is the conversion period of the heat sink and heat source in spring and summer. In winter, the soil heat flux is transferred from the deep layer to the shallow layer, which serves as a heat source to heat the atmosphere. Yue et al. [45] obtained consistent results in the study of semi-arid grassland in the Loess Plateau.

### 3.4. Bowen Ratio Variation

Figure 9 shows the seasonal variation in $\beta$. Overall, $\beta$ of the two stations first decreased, then fluctuated slightly, and finally increased. The seasonal average $\beta$ at Dingxi (6.58) and Qingyang (5.85) was highest in the winter and lowest in the summer (2.51 and 0.71, respectively). The growing season $\beta$ at Dingxi fluctuated around 1, whereas that at Qingyang was < 1. Furthermore, $\beta$ of both stations was low, with multi-year mean values of 2.11 and 0.77, respectively. Precipitation at Dingxi (320 mm) was 1.4 times that of Qingyang

(446 mm) during the same period, and $\beta$ in the semi-arid area was 2.7 times that of the semi-humid area.

**Figure 9.** Seasonal variation in $\beta$ in Dingxi (**a**) and Qingyang (**b**).

### 3.5. Environmental and Ecological Controls on Bowen Ratio

To fully understand the influence of environmental impacts on $\beta$ of farmland ecosystems under dry and wet conditions on the Loess Plateau, the rainfall data from Dingxi and Qingyang Meteorological Stations in the past 30 years were used to divide the growth period into dry and wet months. Months with monthly precipitation greater than the average of the same period over many years were defined as wet months, whereas the opposite pattern indicated dry months. It was found that $\beta$ was mainly affected by $Ts - Ta$, $VPD$, shallow $SWC$, and precipitation. Figure 10 and Table 5 show the relationship between $\beta$ and environmental factors on a monthly scale. The regularity between $\beta$ and $Ts - Ta$ in the semi-humid region ($R^2 = 0.51$) was better than that in the semi-arid region ($R^2 = 0.36$) whether under dry or wet conditions (Table 5). Under drought conditions, the correlation between $VPD$ and $\beta$ in the semi-humid region was more significant ($R^2 = 0.44$), and the coefficient of determination in the semi-arid area was only 0.29. Under humid conditions, the opposite result was observed (Figure 10b). The relationship between effective precipitation (defined as the daily precipitation amount that exceeded 0.5 mm in winter and 4.0 mm in other seasons [28]) and $\beta$ was more significant, and $\beta$ decreased significantly as effective precipitation increased (Figure 10c). As can be seen from the scatter points in Figure 10c, under dry conditions, $\beta$ decreased more rapidly with increased precipitation in the semi-arid area than in the semi-humid area. Figure 10d shows the relationship between $\beta$ and $SWC$; the decrease with $SWC$ was more prominent in semi-arid areas. Under the humid condition, the goodness of fit in the semi-humid region was the highest, reaching 0.63.

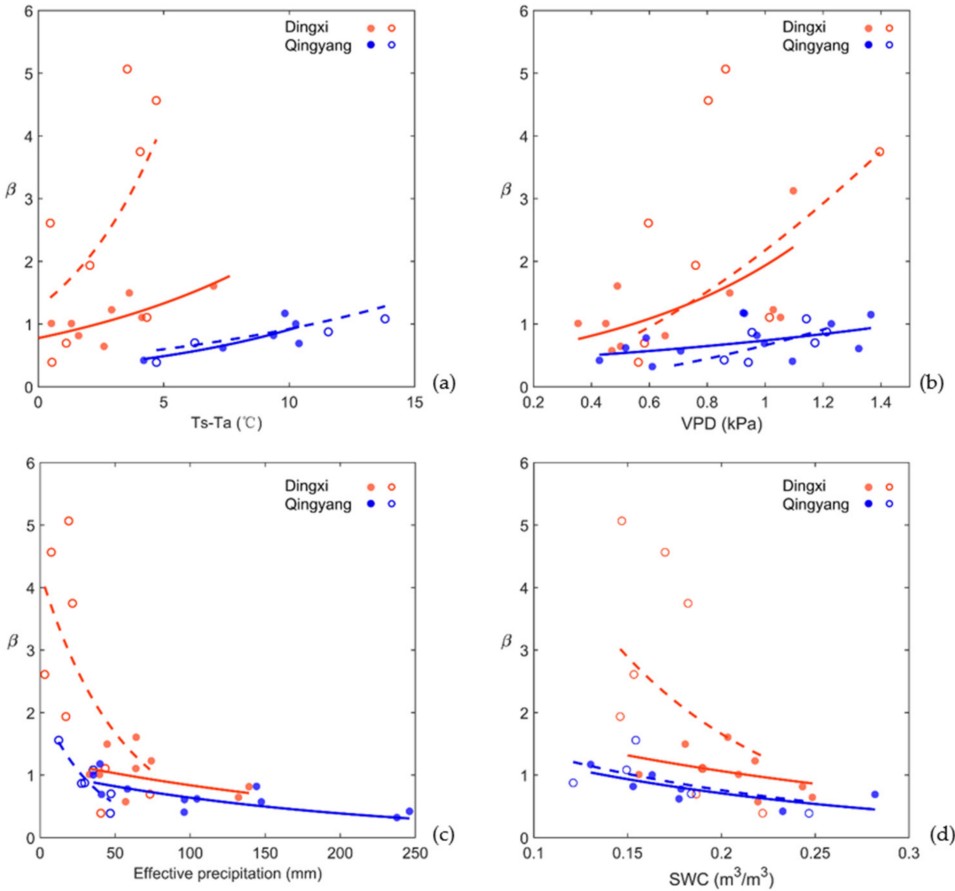

**Figure 10.** Relationship between environmental factors and $\beta$ under different moisture conditions. The solid and hollow dots represent wet and dry conditions, respectively. The red and blue dashed line are the relationships of (**a**) $Ts - Ta$, (**b**) $VPD$, (**c**) effective precipitation, (**d**) $SWC$, and $\beta$ in Dingxi and Qingyang, respectively.

**Table 5.** Statistical relationships between environmental factors and the Bowen ratio under dry and wet conditions.

| Environmental Factors | Equation Parameters | Dingxi | | Qingyang | |
|---|---|---|---|---|---|
| | | **Dry** | **Wet** | **Dry** | **Wet** |
| $Ts - Ta$ (°C) | | $y = ae^{bx}$ | $y = ae^{bx}$ | $y = ae^{bx}$ | $y = ae^{bx}$ |
| | a | 1.27 | 0.78 | 0.39 | 0.26 |
| | b | 0.24 | 0.11 | 0.09 | 0.13 |
| | $R^2$ | 0.36 | 0.59 | 0.51 | 0.58 |
| | $p$ | <0.01 | <0.01 | <0.01 | <0.01 |
| $VPD$ (kPa) | | $y = ax^b$ | $y = ae^{bx}$ | $y = ax^b$ | $y = ae^{bx}$ |
| | a | 2.18 | 0.46 | 0.66 | 0.39 |
| | b | 1.62 | 1.45 | 1.76 | 0.65 |
| | $R^2$ | 0.29 | 0.38 | 0.44 | 0.22 |
| | $p$ | <0.05 | <0.05 | <0.05 | <0.05 |
| Effective precipitation (mm) | | $y = ae^{bx}$ | $y = ae^{bx}$ | $y = ae^{bx}$ | $y = ae^{bx}$ |
| | a | 4.25 | 1.27 | 2.16 | 1.05 |
| | b | −0.02 | −0.004 | −0.03 | −0.01 |
| | $R^2$ | 0.37 | 0.27 | 0.80 | 0.60 |
| | $p$ | <0.01 | <0.01 | <0.01 | <0.01 |
| $SWC$ (m³/m³) | | $y = ae^{bx}$ | $y = ae^{bx}$ | $y = ae^{bx}$ | $y = ax^b$ |
| | a | 15.23 | 2.49 | 2.48 | 0.12 |
| | b | −11.08 | −4.24 | −5.92 | −1.10 |
| | $R^2$ | 0.19 | 0.33 | 0.36 | 0.63 |
| | $p$ | <0.05 | <0.01 | <0.01 | <0.01 |

The relationship between *NDVI–Gs*, *Gs–α*, and *α–β* was determined to explore the influence of ecological factors on the hydrothermal process. *Gs* increased exponentially as the *NDVI* increased. When the *NDVI* was the same in both areas, *Gs* in the semi-arid area was smaller than in the semi-humid area, and the correlation between *NDVI* and *Gs* in the semi-humid area was more significant ($R^2 = 0.57$). The regulation of transpiration by *Gs* is reflected by the Priestley–Taylor coefficient. Figure 11b shows that *α* increased logarithmically with *Gs*. The increasing trend of *α* in semi-humid areas as *Gs* increased is more significant than that in the semi-arid area, with tangent slopes of 0.19 and 0.26, respectively. In addition, *β* decreased exponentially as *α* increased; this trend was more pronounced in the semi-arid area (Figure 11c).

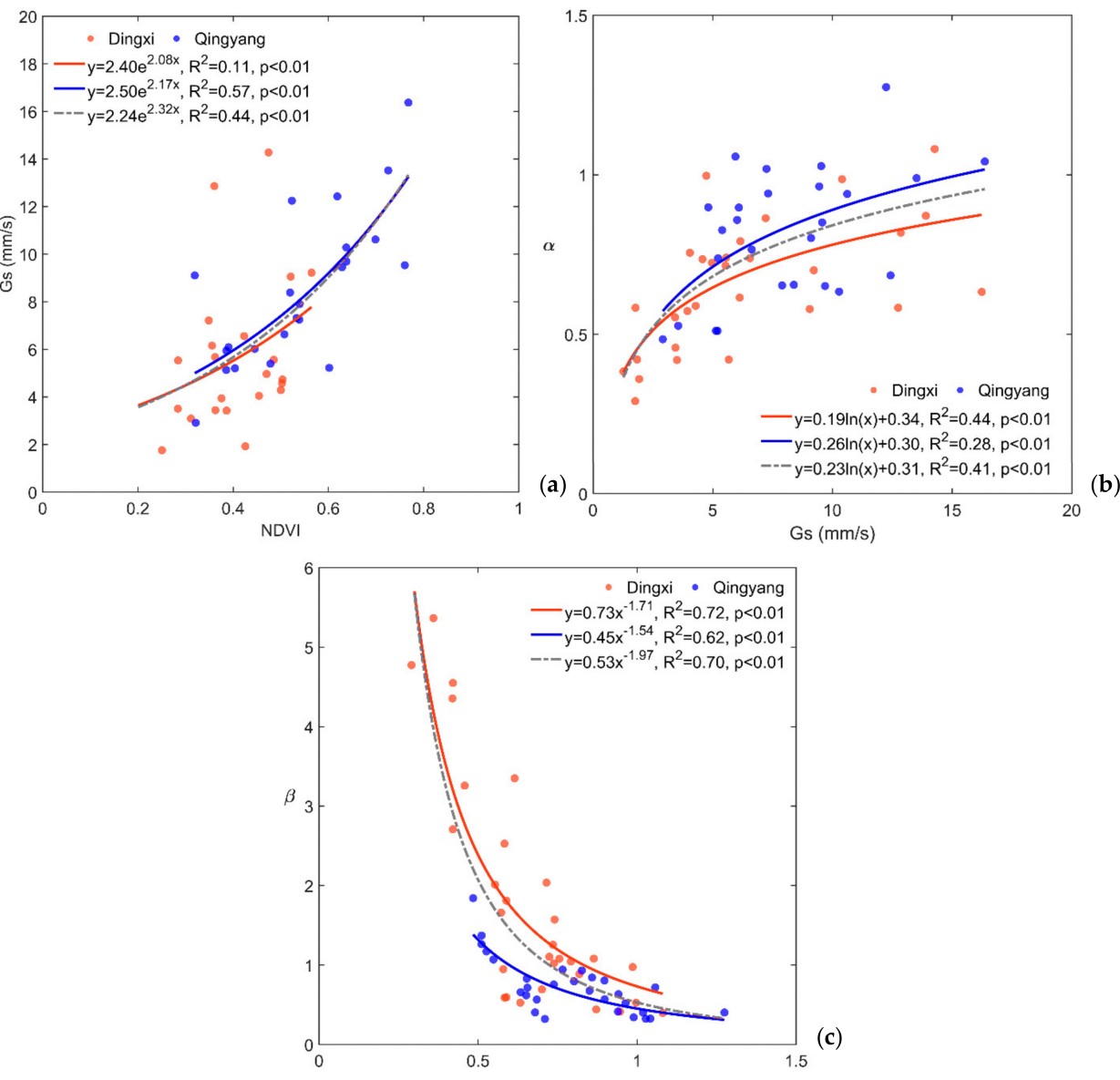

**Figure 11.** Relationship between (**a**) the *NDVI* and *Gs*, (**b**) *Gs* and *α*, (**c**) *α* and *β* on a monthly scale.

We produced path diagrams of the Dingxi and Qingyang stations to further analyze the direct and indirect effects of various influencing factors on *β* (Table 6). On a daily scale, eco-environmental factors at Dingxi and Qingyang explained 60% and 58% of the change in *β*, respectively. From an impact factor perspective, *β* in the semi-arid area was primarily influenced by the direct effect of *NDVI* and the indirect effect of *SWC* (path coefficients of −0.68 and −0.41, respectively). *Ta* and *NDVI* in the semi-humid area were

the most important direct and indirect influencing factors, with contributions of 48% and 17%, respectively. In addition, *SWC* had a more significant overall regulatory effect on $\beta$ in the semi-arid area, with a total path coefficient of $-0.63$. In the semi-humid area, $\beta$ was primarily affected by *Ta*, with a total path coefficient of $-0.53$. It is worth noting that the effect of *Ta* on Bowen ratio in the semi-arid and semi-humid regions was completely opposite. This phenomenon can be explained by the influence of the mechanism found by Zhang et al. [59] in the transition zone affected by the summer monsoon, which is related to *Ta* and land surface evapotranspiration. Under the humid condition, the increase in temperature significantly increases the evapotranspiration, whereas under the drought condition, the increase in temperature decreases the land surface *SWC*, thus inhibiting the surface evapotranspiration. Yue et al. [33] found that the effects of *Ta* in dry and wet years on evapotranspiration were similar by studying the long series of observation data of the semi-arid grassland ecosystem on the Loess Plateau.

**Table 6.** Path coefficient between Bowen ratio and impact factor.

| Site | Correlation Effect | NDVI | Ta | VPD | SWC |
|------|------|------|------|------|------|
| | Direct | $-0.68$ | 0.42 | 0.11 | $-0.21$ |
| Dingxi | Indirect | 0.12 | $-0.36$ | 0.39 | $-0.41$ |
| | Total | $-0.57$ | 0.06 | 0.51 | $-0.63$ |
| | Direct | 0.33 | $-0.90$ | 0.54 | $-0.35$ |
| Qingyang | Indirect | $-0.84$ | 0.37 | $-0.26$ | $-0.01$ |
| | Total | $-0.51$ | $-0.53$ | 0.28 | $-0.36$ |

To comprehensively assess the effects of farmland ecosystem *Gs*, near-ground aerodynamic characteristics, and the local climate background on $\beta$, Cho et al. [24] defined the normalized surface impedance (*Rs\**). Figure 12a shows the relationship between $\beta$ and the monthly mean *Rs\**. There was a significant linear relationship between the $\beta$ and *Rs\** in the farmland ecosystem of the Loess Plateau, with a slope of 0.49. As expected, $\alpha$ decreased more slowly as *Rs\** increased in the semi-arid region than in the semi-humid region due to the growing season. The goodness of fit in the two regions was basically the same ($R^2 = 0.81$).

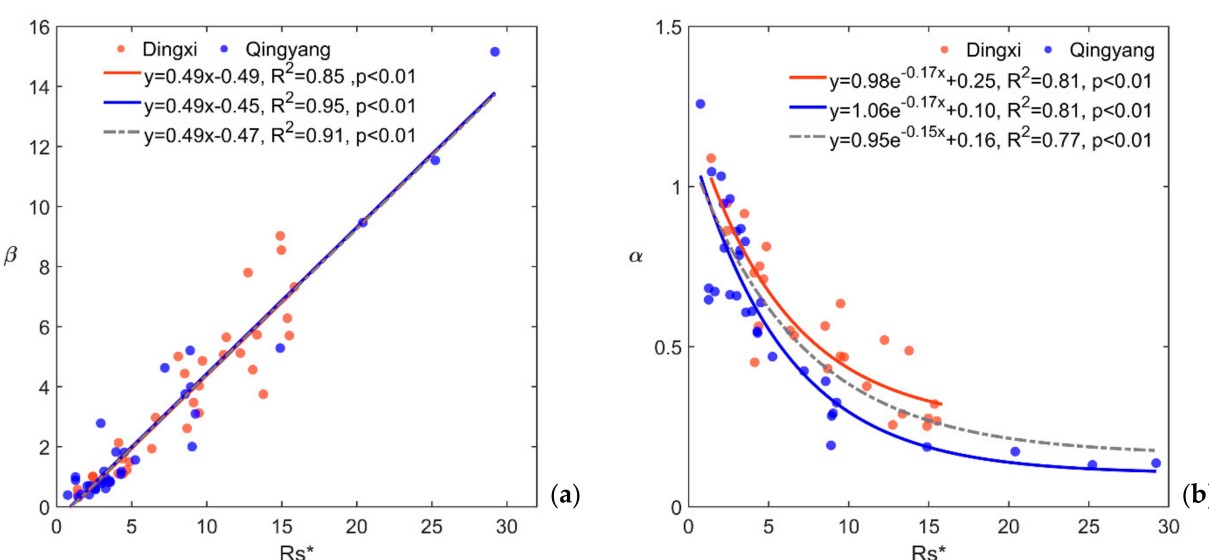

**Figure 12.** Relationships between the normalized surface impedance (*Rs\**) and (**a**) $\beta$ and (**b**) $\alpha$ on a monthly time scale. The gray dotted line in the figure represents the overall trend of the above relationship.

## 4. Discussion

### 4.1. Bowen Ratio Variation

$\beta$ comprehensively reflects the wet and dry conditions of an ecosystem. A $\beta$ of 1 is usually used as the critical value for assessing dry and wet conditions in an ecosystem [58]. Figure 9 indicates that $\beta$ in the Loess Plateau was extremely sensitive to climate fluctuations; semi-arid areas represented by Dingxi were mostly dry. $\beta$ still fluctuated around 1 in summer when precipitation was relatively concentrated. Thus, the land surface ecosystem in the region was largely on the border of the dry and slightly wet areas and was primarily in a state of drought in winter and spring. The fluctuation in $\beta$ at Dingxi in the semi-arid area during the growing season was more significant than that of Qingyang in the semi-humid area. This was due to the low annual total precipitation in this area and the large precipitation variability during the summer monsoon, which led to prominent non-uniformity in the spatio-temporal distribution of soil moisture in the underlying surface and quickly transformed the surface energy distribution. In addition, the precipitation in the semi-humid area was 1.4 times higher than that in the semi-arid area; however, $\beta$ of 2.7 was higher in the semi-arid area than in the semi-humid area. Thus, the $\beta$—which reflects the comprehensive effect of land surface water and heat—is more sensitive than relative precipitation in the transition zone from the semi-humid to semi-arid regions of the Loess Plateau.

Huang et al. [60] found that climate change aggravates drought in semi-arid regions of East Asia and humidifies semi-arid regions of North America. Our findings are comparable with $\beta$ (0.26–0.36) in a farmland ecosystem observed by AmeriFlux [26–28]. The ratio in the farmland ecosystem in the Loess Plateau was approximately 2–3 times higher than that in similar climate regions of North America, indicating that the degree of water stress of the farmland ecosystem in the Loess Plateau is higher than that in North America. This also means that the impact of climate change, especially drought, on the ecosystem of the Loess Plateau in China is greater than that in North America.

### 4.2. Influence of Environmental and Ecological Factors on the Bowen Ratio

Solar radiation [61], rainfall dynamics, and irrigation [60] affect the water potential gradient and surface resistance, and change the soil moisture and evaporation composition of *LE*, thus impacting $\beta$ [3,62]. For the same climatic region, $\beta$ is more sensitive to the changes in $Ts - Ta$ under arid conditions. Further analysis showed that $\beta$ decreased with an increase in *SWC*. This trend was more prominent in semi-arid areas, because the land surface parameters in semi-arid areas were more sensitive to changes in *SWC* [34]. Effective precipitation indirectly affects energy distribution by affecting the content of shallow soil moisture (Figure 10c) [28]. Comparing the relationship between $\beta$ and effective precipitation in semi-arid and semi-humid regions of the Loess Plateau, it was found that the response speed of $\beta$ to effective precipitation is faster than that of the latter. Thus, the change in the effective precipitation in this region had an important impact on the key physical parameters of the land surface ecosystem, and regulated the water and heat exchange of the ecosystem through ecological factors. In this study, the change in $\beta$ with environmental factors was clearer in semi-arid areas than in semi-humid areas; further, the response of $\beta$ to environmental factors was more severe under dry conditions than under humid conditions.

Ecological factors are the principal regulators of *ET* and affect the process of energy distribution through canopy conductance. Evapotranspiration is also closely related to vegetation growth [20,61]. Yue et al. [28] indicated that the *NDVI* regulated transpiration in the ecosystem by affecting *Gs*, which in turn had an important regulatory effect on $\beta$. Differences between climatic regions with various vegetation growth statuses lead to differences in the correlation between the *NDVI* and *Gs*. The ecosystem was less affected by water stress in the semi-humid area than in the semi-arid area, and the growth state of vegetation was better. Because vegetation transpiration is very sensitive to stomatal impedance, the intensity of transpiration increases as *Gs* increases. In theory, when *Gs*

reaches a critical value, $\alpha$ tends to reach equilibrium as $Gs$ increases; thus, the magnitude of evapotranspiration is not limited by $Gs$, and vegetation transpiration reaches a maximum [63]. McHaughton and Spriggs [64] reported the theoretical critical value of ecosystem $Gs$ (12–16 mm/s). Yue et al. [28] found that the critical value of $Gs$ in semi-arid grasslands on the Loess Plateau was 8.2 mm/s. However, this study did not find an obvious $Gs$ threshold in the farmland ecosystem in the Loess Plateau, indicating an ecosystem under water stress. As $Gs$ increased, the $\alpha$ value in the semi-arid area increased less than in the semi-humid area, indicating that ecological factors in the semi-arid area had a more prominent inhibitory effect on evapotranspiration.

*4.3. Biometeorological Controls on the Bowen Ratio*

The water and heat distributions of farmland ecosystems in different climatic regions are restricted by different influencing factors. For semi-humid areas, the energy distribution is affected by the air temperature, which indicates the type of energy constraint. However, the hydrological conditions of $\beta$ are more significant in semi-arid areas and show the characteristics of water restriction. Because the surface of the semi-humid area was relatively humid and the vegetation grew well, the increase in $Ta$ promoted surface evaporation and vegetation transpiration, resulting in the latent heat flux being the dominant energy distribution process. The surface was dry in the semi-arid area, and therefore the lower $SWC$ inhibited soil evaporation and vegetation transpiration, and energy distribution through the sensible heat flux played the leading role in the increase in $\beta$. The effect of $Ta$ on $\beta$ was the opposite in the semi-arid and semi-humid regions. This phenomenon is related to $Ta$ and land surface evapotranspiration [32,59]. Under humid conditions, a temperature increase significantly increases evapotranspiration; under drought conditions, a temperature increase decreases the land surface $SWC$, thus inhibiting surface evapotranspiration.

In this study, $Rs^*$ was used to show the consistencies and differences among different climatic zones of farmland ecosystems on the Loess Plateau. The consistency is reflected in the relationship between $Rs^*$ and $\beta$, which was significantly linear (Figure 12a). Fraedrich et al. [65] found that $Rs^*$ was larger when the surface was dry, that the physiological activity of vegetation was weak, and that the Bowen ratio was accordingly larger. Cho et al. [24] analyzed AmeriFlux observation data and found a positive correlation between $\beta$ and $Rs^*$, which is very sensitive to the physiological vegetation processes. The slope of the regression equation between $\beta$ and $Rs^*$ was 0.21 ($R^2 = 0.65$). Yue et al. [28] found that the slope of $\beta$ and $Rs^*$ in the semi-arid grassland of the Loess Plateau was 0.34 ($R^2 = 0.95$). The slope of the regression equation for the relationship between $\beta$ and $Rs^*$ of the farmland ecosystem in the Loess Plateau was 0.49 ($R^2 = 0.91$), which indicates that the comprehensive influence of the eco-environmental factors of the farmland ecosystem on the Loess Plateau led to a larger $\beta$ than that of the grassland ecosystem in this region. Therefore, the effect of water stress on the farmland ecosystem is more serious than that on the grassland ecosystem. Figure 12b shows the difference between the semi-arid and sub-humid regions and the relationship between $\alpha$ and $Rs^*$. A significant negative correlation between the two was found in the semi-arid farmland ecosystem of the Loess Plateau. For the same $\alpha$ value, the $Rs^*$ of the semi-arid area was greater than that of the semi-humid area, indicating that $Rs^*$ increased as the aridity of the regional climate increased. Additionally, $Rs^*$ had a stronger limiting effect on evapotranspiration in semi-arid areas.

## 5. Conclusions

Under the background of global change, differences in precipitation caused by the duration of the dry and wet seasons in different climate regions are likely to be enhanced. The Loess Plateau in China has both semi-arid and semi-humid climate zones, and it is located in the transitional zone of the East Asian summer monsoon, in which the seasonal variation in precipitation is particularly obvious. Due to the intensity of the monsoon and its advancing northern edge, the land surface processes in the farmland ecosystem of this region display large interannual and seasonal changes. In particular, $\beta$, which characterizes

the intensity of land surface water and heat exchanges, is very sensitive to environmental factors. The precipitation in Qingyang during the growing season was 1.4 times that in Dingxi, but the Bowen ratio in Dingxi was 2.7 times that in Qingyang, indicating that the land surface water and heat exchanges on the Loess Plateau were more sensitive than the precipitation variations. In addition, the farmland ecosystem of the Loess Plateau was more affected by water stress than farmland ecosystems in North America, and the impact of drought on the ecosystem in this region was also greater than in North America. According to the experimental observations, $\beta$ in the Loess Plateau was extremely sensitive to climate fluctuations, and most of the time in the semi-arid region it indicated a dry state. Even during the summer monsoon, the regional land surface ecosystem was subject to water stress. The fluctuation in $\beta$ around 1 indicated that the semi-arid region of the Loess Plateau was on the borderline of dry and slightly wet conditions for long periods. Compared with the semi-humid region of the Loess Plateau, there was less annual total precipitation in the semi-arid region and the precipitation variability was larger during the summer monsoon, which led to obvious fluctuations in the contributions of the latent and sensible heat fluxes to the energy distribution; this was a factor affecting the stability of $\beta$ in the semi-arid region of the Loess Plateau during the summer monsoon.

The main environmental factors affecting $\beta$ of the farmland ecosystem under different dry and wet conditions on the Loess Plateau are $Ts - Ta$, $VPD$, shallow $SWC$, and precipitation. The positive correlation between $\beta$ and $Ts - Ta$ in the Loess Plateau was stronger in the semi-humid region than in the semi-arid region. Under drought conditions, the correlation between $VPD$ and $\beta$ in the semi-humid area was more significant. $\beta$ of the farmland ecosystem in this region decreased with the increase in $SWC$, especially in semi-arid areas, because the land surface water and heat exchanges in semi-arid areas were more sensitive to changes in $SWC$. Ecological factors regulated evapotranspiration through canopy conductance, which then affected $\beta$. The $NDVI$ controlled the transpiration process within the ecosystem by affecting $Gs$, and then played an important role in regulating $\beta$ of the ecosystem. Theoretically, when $Gs$ reaches a critical value, $\alpha$ tends to remain stable with a further increase in $Gs$, and the transpiration of vegetation reaches a maximum at this point. According to our observations, there was no obvious threshold for the farmland ecosystem on the Loess Plateau, but previous studies have found that there is a sensitivity threshold for $Gs$ in the semi-arid grassland on the Loess Plateau, demonstrating that the farmland ecosystem in this region is in a state of water stress. Therefore, from the response of the land surface water and heat exchange processes to the summer monsoon, restoring farmland to grassland in the Loess Plateau may reduce the demand for water evapotranspiration and help to maintain the stability of the regional ecosystem. A path analysis showed that $NDVI$ and $SWC$ had obvious direct and indirect effects on $\beta$ in the semi-arid area, whereas $\beta$ in the semi-humid area was directly and indirectly affected by $Ta$ and $NDVI$. The influence of $Ta$ on $\beta$ in the semi-humid and semi-arid regions had the opposite effect. An increase in temperature in the semi-humid region significantly increased evapotranspiration, whereas an increase in temperature in the semi-arid region decreased the land surface $SWC$ and inhibited surface evapotranspiration.

**Author Contributions:** Conceptualization, Q.Z. and P.Y.; Data curation, X.R.; Formal analysis, X.R.; Funding acquisition, P.Y.; Resources, S.W.; Supervision, Q.Z., P.Y. and J.Y.; Validation, P.Y. and J.Y.; Visualization, X.R.; Writing–original draft, X.R.; Writing–review and editing, X.R. and P.Y. All authors have read and agreed to the published version of the manuscript.

**Funding:** This research was funded by the National Natural Science Foundation of China under grant Nos. U2142208, 41975016, 41705075, and the Basic Science Fund for Creative Research Groups of Gansu Province, grant number 20JR5RA121.

**Informed Consent Statement:** Not applicable.

**Data Availability Statement:** Data not available due to legal restrictions and observation team requirements.

**Acknowledgments:** We are grateful to the NASA Goddard Space Center for providing remote sensing data for this study.

**Conflicts of Interest:** The authors declare no conflict of interest.

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
