# Peer review of "Environmental and Biophysical Effects of the Bowen Ratio over Typical Farmland Ecosystems in the Loess Plateau"

_remotesensing, doi:10.3390/rs14081897_

Round 1

Reviewer 1 Report

The authors were able to improve the overall scientific quality of the manuscript. Just few corrections are needed:

INTRODUCTION: Please, consider in the scientific background of your study also the importyance of climate change and heat transfer in the management of natural resources broadly speaking (i.e.,

Ashraf Vaghefi, S., Mousavi, S.J., Abbaspour, K.C., Srinivasan, R. and Yang, H. (2014), S. Ashraf Vaghefi et al.. Analyses of the impact of climate change on water resources components, drought and wheat yeld in semiarid regions: Karken River Basin in Iran. Hydrol. Process., 28: 2018-2032. https://doi.org/10.1002/hyp.9747.

Mittal, N., Bhave, A.G., Mishra, A. et al. Impact of Human Intervention and Climate Change on Natural Flow Regime. Water Resour Manage 30, 685–699 (2016). https://doi.org/10.1007/s11269-015-1185-6.

Lama, G.F.C., Sadeghifar, T., Azad, M.T., Sihag, P., Kisi, O. 2022. On the Indirect Estimation of Wind Wave Heights over the Southern Coasts of Caspian Sea: A Comparative Analysis. Water 4, 843. https://doi.org/10.3390/w1406084.

Jodar-Abellan, A., Valdes-Abellan, J., Pla, C., Gomariz-Castillo, F. 2019. Impact of land use changes on flash flood prediction using a sub-daily SWAT model in five Mediterranean ungauged watersheds (SE Spain), Sci. Tot. Environ. 657, 1578-1591. https://doi.org/10.1016/j.scitotenv.2018.12.034.

Kalinowska, M.B. Effect of water–air heat transfer on the spread of thermal pollution in rivers. Acta Geophys. 67, 597–619 (2019). https://doi.org/10.1007/s11600-019-00252-y

Milly, P.C.D. and Dunne, Krista A., 2017. A Hydrologic Drying Bias in Water-Resource Impact Analyses of Anthropogenic Climate Change. Journal of the American Water Resources Association (JAWRA) 53( 4): 822838. https://doi.org/10.1111/1752-1688.12538  

RESULTS: Please, just specify the value of the interqaurtile range adpot in the box-plot analysis shown in Fgiure 8b.

After these corrections, the paper is ready for publication.

Author Response

Responses to review comments have been uploaded.

Reviewer 2 Report

Manuscript Recommendation

Review of Remote Sensing manuscript 1667033 –“ Environmental and biophysical effects of the Bowen Ratio over Typical Farmland Ecosystems in the Loess Plateau.

The manuscript evaluates important agronomic and ecological physical parameters that influence energy flow and fluctuation. The research approach is scientifically sound, and results received acceptable statistical evaluation. The approach uses near-ground (tower) sensors for data acquisition. As such, the manuscript addresses physical modeling via an operational process facility, thus it is appropriate for publication in Remote Sensing. The authors have revised an earlier version of this manuscript and addressed the concerns and errors noted in my original my original review. Additional text explanations are incorporated into the document along with an additional figure. The English language usage is good; a final spell check is advisable. The manuscript can be accepted for publication in present form with one minor revision.

The caption in Figure 3 states that the rainfall represents monthly precipitation. The graph, however, is labelled as Rainfall (mm/d).

Author Response

(The authors gave the same response as above.)

Reviewer 3 Report

The authors have addressed the suggested revisions. I think that the modifications have improved the quality of the current manuscript, and I do not have any additional comments.

Author Response

Thanks for your valuable suggestions earlier.

This manuscript is a resubmission of an earlier submission. The following is a list of the peer review reports and author responses from that submission.

Round 1

Reviewer 1 Report

This study presents rich data measured over in Dingxi and Qingyang that were used to conδuct observations of the energy distribution characteristics of semi-arid and semi-humid farmland ecosystems on the Loess Plateau and alongside analysis of the relationships between Physiological and ecological factors based on Bowen ratio. It is well written and organized with proper methods and analysis and can interest many, especially agro-meteorologists. It needs revision, especially since there is a lack of in-depth discussion. Currently, it is more like speculations without enough literature support. This lack of discussion is also reflected by the fact that the cited references appear only before the Results and Discussion session. Finally, it is essential to improve the quality of the figures (higher dpi) and it is suggested to replace figure 11 with a new table including the relevant correlations.

Reviewer 2 Report

This paper investigated the effects of Environmental Factors on the Bowen Ratio of Typical Farmland Ecosystems in the Loess Plateau. I found that this work is important and interesting because it can be an added value to more understating the mass and energy exchanges between the biosphere and atmosphere by developing the land surface modeling. The paper is well written and the methodology is well presented, however my concern is that the paper doesn’t’ any value in the remote sensing domain. It can be accepted after my comments and suggestions below are fully considered.

Introduction:

Line 61 : add a reference ’AmeriFlux observations have ….. is  between 0.26 and 1.3’

Line 67 : ‘Precipitation is the most important driving factor in this process’ Irrigation can be also an important driving factor ????:

Materials and Methods :

Line 107 : ‘The mean annual temperature and precipitation are 6.7 °C and 386 mm’ add about after are.

Line 108 : same remark as in line 107, ‘’The water surface evaporation is about 1400 mm’’ 

  Figure 1 : the star of Dx site is not clear with the red background.Table1 : how did you measure soil temperature at 0cm ? in practice it’s not possible to put sensor at 0 cm.  

Line 136 : ‘’after excluding the outliers and precipitation period data, missing data for periods of less than 6 h were linearly interpolated’’. This may be generates some eroors in the fluxes values especially during the pick of LE and H ? How can can this method affects the EBC, can you give the error of the uncertainty of this method ?

Energy balance :

Line 145 : you need to state why you neglected the other terms of the equation auch as the rate of change of heat storage in the above-ground biomass and the energy flux density associated with the CO 2 flux (through photosynthesis and respiration)

Soil heat flux correction section :

Line 165 : Soil temperatures at depths of  2 is not measured as mentionned in table 1

  Line 212 : inaddition -> in addition

Environmental factor variations section :

-          Please describes also the variation of the Priestley–Taylor coefficient which is presented in the figure 2-          Add SWC in the legende near to the depths or add the depths in the title near to  SWC is the soil water content in brackets.

Energy balance characteristics :

‘’This was due to the strong solar …………….. At night, due to the low wind speed and insufficient turbulent mixing, the energy closure was low ‘’. May be this can be more related to the increased uncertainty in energy fluxes during the stable conditions ?

Table 2 k is the slope ????

Can you explain why the OLS method improves the Energy balance closure  at whole day compared to EBR

Figure 5 : G also plays an imprortant rule in the partition of Rn especially during summer and autumn; add some disucsion about it. The same remark for the figure 6, any word about G.

Delete the lettres a, b,c and d from the figure.

Environmental and ecological Bowen ratio controls section :

Line 371: explain why theory gives a negative correlation and your experiment gives positive one??? This is in opposite to what you stated in the ‘’Influence of environmental and ecological factors on the Bowen ratio section’’.  See line 458 : ‘’Ta is theoretically negatively correlated with the Bowen ratio [14]. The experimental results showed that the relationship in the Loess Plateau agreed with the theoretical results’’

Reviewer 3 Report

Manuscript Recommendation

Review of Remote Sensing manuscript 1564380 –“ Effects of Environmental Factors on the Bowen Ratio of Typical Farmland Ecosystems in the Loess Plateau.

The manuscript evaluates important agronomic and ecological physical parameters that influence energy flow and fluctuation. The research approach is scientifically sound, and results received acceptable statistical evaluation. The approach uses near-ground (tower) sensors for data acquisition. As such, the manuscript addresses physical modeling via an operational process facility, thus it is appropriate for publication in Remote Sensing. Minor revision is needed prior to publication. Specifically, additional detail is needed to define the study period and how unbalanced data was compiled and analyzed across the two study sites. Detailed comments follow that outline the key concerns. The manuscript abstract is rewritten to help clarify the content. The rewrite is intended as a guide. Authors should evaluate the rewrite to assure the original content has not been misstated.

Rewrite

Abstract: The Bowen ratio (Br) comprehensively reflects physical characteristics of the land-surface climate. In this study, eddy covariance systems installed at Dingxi and Qingyang were used to conduct energy distribution measurements and observations characteristic of semi-arid and semi-humid farmland ecosystems on the China Loess Plateau. We studied mechanisms by which eco-environmental factors influence Br. Additionally, we investigated responses of physiological and ecological factors to water and heat exchange under seasonally dry and wet conditions within each farmland ecosystem. Our results showed that sensible heat flux in the semi-arid farmland was the main consumer of available energy. In the semi-humid area, latent heat flux in summer had the dominant role in energy distribution (mean  Br 0.71). The Br in the semi-arid region was 1.5 times higher than that in the  semi-humid region during the growing season. Br increased with an increase in the vapor pressure deficit (VPD), and decreased significantly with an increase in air temperature, precipitation, and soil  moisture. The change in Br with environmental factors was more clear-cut  in semi-arid areas  than in semi-humid areas. The Priestley-Taylor coefficient (α) and Br satisfied a power function law in the growing season. There was a strong correlation between temperature and Br with the coefficients of determination for semi-humid and semi-arid areas being 0.35 and 0.72, respectively. Br decreased with an increase in the normalized difference  vegetative index (NDVI), with this phenomenon being more obvious in the semi-humid zone (R2=0.40). Br responded more rapidly to NDVI in the semi-arid area than in the semi-humid area. There was a negative exponential relationship between canopy stomatal conductance (Gs) and Br, which displayed a stronger declining trend with the increase of Gs in the semi-arid area than in the semi-humid area. This study provides an important reference for the determination of land-surface characteristics of semi-arid and semi-humid farmland ecosystems on the Loess Plateau and for improving parameterization of land-surface processes.

General comment

The manuscript uses the term Bowen ratio extensively. I suggest that the authors define an abbreviated symbol for Bowen ratio (e.g., Br) to simplify the verbiage.

Abstract

Lines 26-28.

What is the strong correlation that is described? The value 0.72 suggests that the correlation referred to is between temperature and Bowen ratio. The stated correlation value 0.62, however, does occur in the data. Should the value be 0.35 as given in Table 5 for temperature (Ta) vs Br at Dingxi. The sentence needs correction and/or clarification. 

Page 7, Figure 2

Page 12 Figure 5

These figures present environmental factor data for the Dingxi site during July 2016 through July 2019.

The Qingyang site shows data for parts of 2012 through about March of 2016, no data for 2017, and data for March 2018 through July 2019.

What exactly is the period of study? The period of study needs better definition.

Why was data not uniformly collected at both sites? Need explanation.

How was the unbalanced data compared, compiled, and statistically analyzed to account for variation across several years when data is not available at both sites?

These details need greater explanation and detail.

Page 7

Figure 2

The legend shows a line symbol for Rainfall. The graphs for both Dingxi and Qingyang show no line for Rainfall. The rainfall data needs to be plotted.

Pag12

Line375

The R2 values are given in Table 5 not in Figure 9a as stated.

Reviewer 4 Report

The article is well written, and the topic is very stimulating in all fields of geosciences. Here you can find my comments:

OVERALL: Please, replace all figures with HD images, and check all sections to fit the template of the Journal.

INTRODUCTION: Please, consider in the scientific bakgroundsome example of influence of biomass vegetation phenology on the response of vegetated ecosystems (i.e., "Lama, G.F.C.; Crimaldi, M.; Pasquino, V.; Padulano, R.; Chirico, G.B. Bulk Drag Predictions of Riparian Arundo donax Stands through UAV-Acquired Multispectral Images. Water 2021, 13, 1333, https://doi.org/10.3390/w13101333).

Vélez-Nicolás, M.; García-López, S.; Barbero, L.; Ruiz-Ortiz, V.; Sánchez-Bellón, Á. Applications of Unmanned Aerial Systems (UASs) in Hydrology: A Review. Remote Sens. 2021, 13, 1359. https://doi.org/10.3390/rs13071359.

METHODOLOGY: Please use some figures to improve the clarity of the methods proposed in all subsections. This is crucial for guiding the reader carefully. Otherwise, the scientific quality of your work is poor.

I will reconsider the article after major revision.